# A New Frameshift Mutation of *PTEN* Gene Associated with Cowden Syndrome—Case Report and Brief Review of the Literature

**DOI:** 10.3390/genes14101909

**Published:** 2023-10-05

**Authors:** Claudia Maria Jurca, Ovidiu Frățilă, Tiberia Iliaș, Aurora Jurca, Andreea Cătana, Corina Moisa, Alexandru Daniel Jurca

**Affiliations:** 1Department of Preclinical Disciplines, Faculty of Medicine and Pharmacy, University of Oradea, 410081 Oradea, Romania; claudiajurca70@yahoo.com (C.M.J.); alexandrudanjurca@yahoo.com (A.D.J.); 2Regional Center of Medical Genetics Bihor, County Emergency Clinical Hospital Oradea (Part of ERN-ITHACA), 410469 Oradea, Romania; 3Department of Medical Disciplines, Faculty of Medicine and Pharmacy, University of Oradea, 410081 Oradea, Romania; ovidiufr@yahoo.co.uk; 4Faculty of Medicine and Pharmacy, University of Oradea, 410081 Oradea, Romania; aurora.jurca@yahoo.com; 5Faculty of Medicine, University of Medicine and Pharmacy “Iuliu Hatieganu”, 400012 Cluj Napoca, Romania; 6Department of Pharmacy Disciplines, Faculty of Medicine and Pharmacy, University of Oradea, 410081 Oradea, Romania; corinamoisa@hotmail.com

**Keywords:** Cowden syndrome, *PTEN* gene, intestinal polyps, hamartomas

## Abstract

Cowden syndrome (CS) is a rare disease that was first described in 1963 and later included in the large group of genodermatoses. It is the most common syndrome among the *PTEN*-associated hamartomatous tumor syndromes (PHTS). CS has an autosomal dominant inheritance pattern, with increased penetrance and variable expressivity, making early diagnosis difficult. Mutations in the *PTEN* gene (phosphatase and TENsin homolog) are involved in its pathogenesis, involving many organs and systems originating in the three embryonic layers (ectodermum, endodermum, and mesodermum). The consequence is the development of hamartomatous lesions in various organs (brain, intestines, thyroid, oropharyngeal cavity, colon, rectum, etc.). Multiple intestinal polyps are common in patients with CS, being identified in over 95% of patients undergoing colonoscopy. The authors describe the case of a patient who presented the first signs of the disease at 3 ½ years (tonsil polyp) but was diagnosed only at the age of 20 following a colonoscopy that revealed hundreds of intestinal polyps, suggesting further molecular testing. A heterozygous frameshift mutation was identified in the *PTEN* gene, classified as a potentially pathogenic variant (c.762del.p(Val255*)). The authors present this case to highlight the path taken by the patient from the first symptoms to the diagnosis and to emphasize the clinical aspects of this mutational variant that have still not been identified in other patients with this syndrome.

## 1. Introduction

Cowden syndrome (MIM≠158350) is a monogenic disease with autosomal dominant transmission; it is part of the PHTS group (*PTEN*-associated hamartoma tumor syndrome), which also includes Bannayan–Riley–Ruvalcaba syndrome (BRRS), *PTEN*-related Proteus syndrome (PS), and *PTEN*-related Proteus-like syndrome [1]. In 1963, Lloyd and Denis described this condition and named it after their patient, Rachel Cowden. She presented a complex phenotype that included craniofacial dysmorphism (macrocephaly, mandibular hypoplasia, microstomia, ogival palatine vault), papillomatosis of the tongue and oropharynx, skeletal anomalies, multiple thyroid adenomas, and fibrous cysts in the breasts [2]. CS is the most common syndrome in this group, and it is usually diagnosed in adulthood; in contrast to BRRS, which has an earlier onset, these patients frequently present symptoms from the autistic spectrum [3], one of the first signs that alarm and direct parents to a specialist medical assessment. The gene responsible for the occurrence of this syndrome is *PTEN*, which is located on the long arm of chromosome 10 (10q23) and was identified and located by Nelen in 1996 [4]. It is a gene with dual action; on the one hand, it acts as a lipid phosphatase, and on the other hand, it acts as a protein phosphatase [5]. The molecular diagnosis of pathogenic variants is critical, although gene mutations are not identified in over 20% of cases [6]. Although it has a dominant inheritance pattern, most cases are isolated. The syndrome has a prevalence of 1:200,000, predominantly reported in the Caucasian population (96%), with the sex ratio slightly favoring the female sex (women are affected in 60% of cases). Only 500 cases are currently reported in the literature [7]. Due to the variable expressivity, it is often diagnosed late. The specific signs of the syndrome are often categorized as belonging to other pathologies. The average age of diagnosis is 39 years [8]. Gastrointestinal involvement is common in CS, and the appearance of intestinal polyps is the pathognomonic sign; polyps are present in 85% of affected people and have various histologies: hamartomatous, inflammatory, adenomatous, hyperplastic, ganglioneuromatous [9].

In 1996, the group working for the International Consensus for CS established the major and minor criteria for diagnosing patients with CS [10]. Subsequently, the criteria underwent a critical revision, especially at the level of the major criteria. The US National Comprehensive Cancer Network later accepted this revision [11]. The diagnostic criteria are summarized in Table 1**.**

This article emphasizes the importance of the early diagnosis of Cowden syndrome to facilitate better monitoring of its clinical evolution and rapid therapeutic intervention. The case presented by the authors had a whole odyssey, from the first sign of the disease (the peritonsillar polyp) to the diagnosis almost 15 years later. At that time, he (the patient) already had hundreds of intestinal polyps. The mutational variant identified in the presented case has not been reported in the specialized literature.

## 2. Materials and Methods

### 2.1. Case Report

The authors present the case of a 20-year-old patient based on the records of CRGM Bihor from April 2023; the patient was referred by a gastroenterology medical specialist for genetic advice. He is the second child in the family born from pregnancy with physiological evolution, spontaneous birth, cranial presentation at 40 W with a weight of 4600 g, and normal neonatal evolution. **Family history**: organized family, no consanguineous relationship. The paternal grandfather died of colon cancer at 74. The maternal grandmother had a thyroidectomy (2006) for multinodular goiter and a left nephrectomy. **Past medical history**: From the age of 3 ½ years, repeated referrals to different medical specialists (all tumor-related) as follows: 2007 (3 ½ years) a tonsillar papilloma surgically removed at the age of 4; 2012 at 9 years, plantar tumor (diagnosed as a plantar hemangioma) surgically excised; 2015 at 12 years of age, an adenomatous lesion of the left thyroid lobe was identified, and a left thyroid lobectomy was performed (the patients was also receiving dermatological care for juvenile acne); 2018 at 15 years old, an ultrasound identified a nodular lesion in the right thyroid lobe that is still under follow-up; 2023 at 20 years of age, the exacerbation of dyspeptic syndrome resulting it the patient becoming unresponsive to medication with digestive enzymes and antispasmodics, leading to gastroenterology consultation followed by a colonoscopy that revealed multiple polypomatous lesions with a suspected genetic-associated polyposic syndrome—a medical genetic assessment was recommended. Furthermore, in 2023, a dermatological consultation revealed a papular lesion on the forehead, with the clinical appearance of trichilemmoma on the forehead and facial acne; a biopsy of the papular formation was recommended.

### 2.2. Laboratory Investigations

Laboratory tests were focused on the assessment of thyroid hormones and hematological parameters.

### 2.3. Molecular Investigations

The total genomic DNA was extracted from the biological sample using a bead-based method. After the fluorometric assessment of DNA quantity, the qualified genomic DNA sample was randomly fragmented using non-contact, isothermal sonochemistry processing. Molecular tests were performed at Blueprint Genetics. The patient was tested using next-generation sequencing (NGS) with a multi-gene panel of 43 genes that were selected based on their associations (according to what has been reported in the medical literature) with Hereditary Gastrointestinal Cancer Panel Plus [13], which includes sequence analysis and copy number variation analysis of the following genes: *APC*, *ATM*, *AXIN2*, *BLM*, *BMPR1A**, *BRCA1**, *BRCA2*, *BUB1B*, *CDH1*, *CDKN2A*, *EPCAM*, *FANCC*, *GALNT12*, *GREM1*, *KIT*, *MEN1*, *MLH1*, *MLH3*, *MSH2*, *MSH3*, *MSH6*, *MUTYH*, *NF1**, *NTHL1*, *PALB2*, *PDGFRA*#, *PMS2**, *POLD1*, *POLE*, *PTEN**, *RHBDF2*, *RPS20*, *SDHB*, *SDHC*, *SDHD*#, *SMAD4*, *SMARCB1*, *STK11*, *TMEM127*, *TP53*, *TSC1*, *TSC2,* and *VHL*. **The sequencing library** was prepared by ligating sequencing adapters to both ends of the DNA fragments. Sequencing libraries were size-selected using a bead-based method to ensure optimal template size and subsequently amplified via polymerase chain reaction (PCR). Regions of interest (exons and intronic targets) were targeted using a hybridization-based target capture method. The quality of the completed sequencing library was controlled by ensuring the correct template size and quantity and eliminating the presence of leftover primers and adapter–adapter dimers. Readily available sequencing libraries that passed the quality control stage were sequenced using Illumina’s sequencing-by-synthesis method using paired-end sequencing (150 by 150 bases). **Bioinformatics and quality control**: Burrows–Wheeler Aligner software was used for read alignment. The variant classification follows the modified Blueprint Genetics Variant Classification Schemes from the ACMG guideline 2015. The patient’s sample was subjected to thorough quality control measures, including contamination and sample mix-up assessments. Copy number variations (CNVs) were detected from the sequence analysis data using a proprietary bioinformatics pipeline. The expected sequencing depth was obtained by using other samples processed in the same sequence analysis as a guiding reference. The sequence data were adjusted to account for the effects of varying guanine and cytosine content. All available evidence of the identified variants was compared to the classification criteria. Sequence variants classified as pathogenic, likely pathogenic, and variants of uncertain significance (VUS) were confirmed using bi-directional Sanger sequencing when they did not meet our stringent NGS quality metrics for a confirmed positive call. The molecular analysis revealed the heterozygous c.762del, p.(Val255*) mutation in the *PTEN* gene, a likely pathogenic frameshift mutation. Written consent was obtained in order to include case details and all images.

## 3. Results

### 3.1. Clinical Evaluation of the Patient

Phenotypical traits: height 177 cm (75th percentile), weight 97 kg (over 95th percentile); craniofacial dysmorphism (macrocephaly with a head circumference of 64 cm) (over 99th percentile), round face, multiple folliculitis lesions, a 2/3 mm scar affirmatively after the excision of a comedo and a papillomatous lesion (trichilemmoma) on the forehead; anterior thoracic, especially posterior, shows multiple pustular lesions—suprasternal keloid scar after surgical removal of the left thyroid lobe. (Figure 1 and Figure 2).

### 3.2. Laboratory Investigations

Our laboratory analyses focused mainly on thyroid hormones. The TSH level was slightly increased in the year of diagnosis of the left thyroid follicular nodules, with a value of 4.1 mIU/L (reference values: 0.40–4.00 mIU/L). After the left thyroid lobectomy, the TSH and thyroid hormone levels remained normal. 

### 3.3. Interdisciplinary Consultations

Table 2 summarizes the interdisciplinary consultations.

### 3.4. Molecular Investigations

Extensive germline testing (43 genes) to assess predisposition to hereditary gastrointestinal and colorectal syndromes was positive. The c.762del.p(Val255*) heterozygous mutation was identified in the *PTEN* gene. This variant is absent in gnomAD, an extensive reference population database (*n* > 120,000 exomes and >15,000 genomes) that aims to exclude individuals with severe pediatric diseases. This variant generates a frameshift in exon 7 (of 9 exons), resulting in a premature stop codon. This is predicted to lead to a loss of normal protein function, either through protein truncation or nonsense-mediated mRNA decay. Loss of function is an established disease mechanism in this gene (HGMD). In silico prediction algorithms (POLYPHEN, SIFT, MUTTASTER) support the pathogenicity of the identified variant, thereby also supporting the negative effect on the phenotype. To our knowledge, this variant has not been reported in the medical literature or in disease-related variation databases.

## 4. Discussion

### 4.1. Clinical Aspects

#### 4.1.1. Craniofacial Dysmorphism

Patients with CS present macrocephaly, broad forehead with frontal bossing, a widened nose, mandibular hypoplasia, microstomia, and ogival palatine vault. The presented case associates macrocephaly with a cranial circumference of 64 cm, a prominent forehead, anteverted ears, and a wide-rooted nose. Macrocephaly is most commonly reported to be identified in 80–100% of patients with PHTS [14,15]. Macrocephaly is defined as an occipitofrontal circumference (OFC) 2 SD or higher than the mean for age, gender, and ethnicity measured over the greatest frontal circumference. Macrocephaly is found in more than 100 OMIM syndromic or non-syndromic entities like Fragile X, Nevoid Basal Cell Carcinoma (NBCCS), and Sotos syndrome, and requires complex clinical assessment and further molecular analysis for diagnosis. 

#### 4.1.2. Thyroid Involvement 

Thyroid involvement is a constant presentation in CS. There is a wide range of manifestations, from benign tumor lesions (adenomatous follicles, nodular goiter) to malignant or autoimmune anomalies (Hashimoto or lymphocytic thyroiditis) [16,17]. Thyroid damage occurs in early childhood and can be diagnosed via thyroid ultrasounds [18]. Thyroid function should be constantly monitored in terms of the risk of malignant transformation (TSH, Tg, TgAc) and to identify functional changes in this gland [19]. Clinicians should consider that the presence of thyroid adenomatous follicles could be an important indication of this disease. In the case of our patient, thyroid nodules were diagnosed at the age of 12 clinically, sonographically, and paraclinically via assessing whether their TSH values were outside of the reference values. At the age of 15, a left lobectomy was performed. Two years after the surgical intervention, the patient developed a nodular lesion in the right thyroid lobe.

#### 4.1.3. Lymphoid Involvement

Autoimmune-related phenotypes and lymphoid hyperplasia can be observed in 40% of PHTS patients [20]. These patients show dysregulated immune function with lymphopenia, CD4+ T cell depletion, and changes in T and B lymphocyte subsets. Tonsillar hypertrophy following the expansion of lymphoid tissue has been described in CD and PHTS [21,22,23]. This can being about varying degrees of hypertrophy, from small asymptomatic sizes to large sizes that cause functional respiratory disorders (difficult oral breathing) and sleep apnea, more commonly found in BRRS [24]. The appearance of tonsillar nodules can also cause upper airway obstruction [25,26]. In the case presented by the authors, the tonsillar polyp was the first warning sign of the disease. It appeared at the age of 3 ½ years and was surgically excised. The postoperative evolution was uneventful. Six months after the resection of the tonsillar papillomatous formation, the boy showed an increase in the size of the papilloma, causing severe functional disorders, including difficult oral breathing, having a permanently open mouth, and eating disorders (feeding only with liquid foods), meaning that emergency tonsillectomy was required at the age of 4.

#### 4.1.4. Gastrointestinal Manifestations

Intestinal polyps represent the most essential and pervasive feature of patients with CS and other PHTS patients. Additionally, >90% of *PTEN* mutation carriers who had a colonoscopy had colorectal polyps, typically with mixed histologies [27]. Polyps can be present at any level of the digestive tract, starting from the stomach to the level of the anus; from an anatomopathological point of view, polyps can be inflammatory, hamartomatous, adenomatous, or ganglioneuromatous [27]. Their presence is often detected due to rectal bleeding or intestinal blockage through intussusceptions [28]. There is a 9 to 16% lifetime risk for malignant transformations for polyps, with the average age of diagnosis being between 44 and 48 years [29]. In patients with mutations in the *PTEN* gene, the prevalence of colorectal polyps can reach 90% [30,31]. Our patient, at the age of 17 ½ years, began to present anal bleeding, initially interpreted as being due to hemorrhoids. In May 2023, following a lower digestive endoscopy and a colonoscopy, more than 100 intestinal polyps were identified from the ileum to the anus. A biopsy was performed for five polyps, four of which had inflammatory histopathological characteristics, and one adenoma. 

#### 4.1.5. Skin Changes

Characteristic skin lesions identified in CS include trichilemmomas, acral keratosis, mucocutaneous neuromas, oral papillomas, and macular pigmentation of the penis gland, all of which are often the first clues for clinical diagnosis [32]. The most common changes that can occur are trichilemmomas-type lesions that present more frequently in the ears, forehead, and perinasal area. However, they can appear in any area of the body. Trichilemmomas are benign tumors originating from the outer cells of hair follicles; usually asymptomatic, they appear as solitary or multiple soft, smooth, skin-colored papules. Their biopsy is usually necessary for diagnosis. Palmoplantar keratoses, gingival papilomas, and hemangiomas may also occur [33,34]. A tongue with a scrotal appearance, lingual nodule, ogival palatine vault, and cutaneous neuromas can also be spotted [35]. Identifying skin lesions in childhood can significantly impact the disease’s evolution, being an alarm sign for a hereditary tumor syndrome, thus allowing its rapid identification and the application of early cancer screening [36]. The presented case was located on the upper part of the forehead and chest (multiple pustular lesions and a skin-colored papular lesion with the appearance of trichilemmomas); the dermatologist recommended the biopsy of this lesion, which also revealed a surgically excised plantar hemangioma.

#### 4.1.6. Cancer Risk

Patients with *PTEN* germline variants have a higher risk for malignant events compared to the general population [37]. A recent literature data review reported higher risks for breast cancer (67% to 85%), endometrium cancer (19% to 28%), thyroid cancer (6% to 38%), renal cancer (2% to 24%), colorectal cancer (9% to 32%), and melanoma (0% to 6%), with a Cumulative Lifetime Cancer Risks (CLTRs) of 81–90% and a median age at diagnosis of 36 years [38]. Another extensive study revealed a higher incidence of cancer in *PTEN* carriers, providing a guide for risk-management strategies with enhanced surveillance approaches to improve clinical outcomes regardless of their initial clinical presentation [39]. Breast cancer is the most common malignancy in *PTEN* female carriers; the tumors are more commonly triple negative (loss of progesterone, estrogen, and HER2 expression), are more aggressive, and could affect both breasts [40]. Despite being a rare occurrence, breast cancer has also been reported in male *PTEN* carriers [41].

#### 4.1.7. Neurodevelopmental Anomalies

Neurodevelopmental anomalies have also been reported in *PTEN* carriers. Along with macrocephaly, research in the literature indicates potentially increased rates of developmental delay and autism spectrum disorder (ASD) and other behavioral and psychological manifestations [42]. A representative percentage of approximately 25% of people with germline *PTEN* mutations also have characteristic criteria of ASD [43]. As mentioned above, macrocephaly is an endophenotipocal trait in CS; interestingly, these patients have increased brain mass and white matter volumes [44] that might be linked to various intellectual disabilities and distinctive cognitive profiles (delayed speech, poor working memory, and processing speed) [45]. Other structural anomalies of the central nervous system, such as meningiomas, arteriovenous malformations, large perivascular spaces, cortical dysplasia, and gray matter heterotopias, have also been reported [46]. In addition to being pathognomonic for CS, Lhermitte–Duclos disease is diagnosed in adults and is characterized by a hamartomatous overgrowth of the cerebellum and large neuronal cells with a “tiger-striped” appearance expanding into the granular and molecular layers of the cerebellar cortex upon imagining assessment [47]. 

### 4.2. Genetics

#### The *PTEN* Gene

The *PTEN* gene is located on the long arm of chromosome 10 (10q23) and consists of 9 exons that encode a protein consisting of 403 amino acids [48]. Germline mutations occur most frequently in exons 5,7,8 and are found more frequently in cancers under the PHTS umbrella [49]. On the other hand, somatic mutations also occur in other tumors, such as prostate cancer, glioblastoma multiform, and even uncommon localizations [50]. More than 300 pathogenic variants in the *PTEN* gene have been described [51]. The *PTEN* gene codes for a dual-activity phosphatase with a tumor suppressor effect that, under normal conditions, inhibits the phosphatidylinositol 3-kinase (PI3K) signaling pathway through its lipid phosphatase activity but simultaneously negatively regulates the MAPK pathway through its protein phosphatase activity. *PTEN* activity occurs at both the cytoplasmic and nuclear levels [52].

**Cytoplasmic activity**. As a lipid phosphatase, *PTEN* plays a negative feedback role in the PIP3/AKT/mTOR signaling pathway. Typically, the cascade is initiated by binding signal molecules to growth factor tyrosine kinase (RTK) receptors. Activated growth factors will recruit and activate phosphatidylinositol 3-kinase. Activated PI3K will downstream phosphorylate phosphatidylinositol 4,5-diphosphate (PIP2) to the second messenger phosphatidylinositol 3,4,5-trisphosphate (PiP3), which will activate serine/threonine protein kinase B (AKT) both directly and by recruiting lipoamide pyruvate dehydrogenase kinase isozyme 1 (PDK1). In turn, AKT will inhibit the tuberous sclerosis complex (TSC), inhibiting the mTOR (mechanistic target of rapamycin) complex [53,54,55,56]. Through its lipid phosphatase function, *PTEN* dephosphorylates the secondary messenger molecule PIP3 to PIP2 (basically causes PIP3 to convert back to PIP2), effectively inhibiting AKT activation; consequently, the mTOR complex will super-activate, causing cell proliferation, cell transformation, and tumorigenesis. *PTEN* has a negative feedback role in the RAS/MAPK pathway by inhibiting the adapter protein Grb2 (Growth factor receptor bound protein 2) through its protein phosphatase activity. Grb2 connects growth factor receptors and the Ras-MAPK signaling pathway through SH domains in its structure (it has one SH2 domain and two SH3 domains). Through the SH2 domain, it attaches to the activated RTK receptor, and through the SH3 domains, it interacts with one end of the **SOS** (son-of-sevenless) protein; the other end of the SOS protein binds to a domain called Ras-GEF, through which it interacts with the Ras protein. Later, RAS will activate the RAF molecule downstream, phosphorylating MERK 1 and 2 and activating ERK1 and 2; thus, it plays an essential role in cell growth and cell proliferation [48,57].

At the nuclear level, *PTEN* plays a significant role in genomic stability, chromosomal architecture maintenance, and cell cycle control [58].

### 4.3. Genotype–Phenotype Correlation

Cowden syndrome and BRRS are two conditions in this group that occur due to mutations in this gene. Despite the fact that, initially, the two syndromes were considered two different entities, nowadays, the two represent a single disease but with different penetrance and variable expressivity caused by the same mutation in the PTEN gene [59].

The correlation between genotypes and phenotypes needs to be better defined to personalize the screening tests. However, some studies emphasize that missenses are rare in thyroid neoplasms with mutational variants, and frameshifts are much more common [60,61]. Nonsense variants would be associated more frequently with colorectal cancer, missense variants would be associated more frequently with autism spectrum disorders, and those in the promoter region would be associated more frequently with breast cancer [62]. Other studies, such as [63], have demonstrated no genotype–phenotype correlation; however, it should be noted that their sample included only 13 cases with PHTS. Estimating the risk for organ-specific cancer, as with other hereditary cancers, is challenging, if not impossible; therefore, it is relative to determining who and what type of cancer will develop during their lifetime. Without a clear genotype–phenotype correlation, it is suitable for patients diagnosed at a young age with a pathogenic or likely pathogenic variant to be monitored according to the recommendations made by international cancer surveillance guidelines [64]. Since the defect identified in the patient was not reported in other patients with *PTEN*-related syndromes, it is impossible to make genotype-to-genotype associations; thus, we had to make assumptions about the phenotype based on a defect with the same molecular characteristics. New findings suggest that frameshift mutations are more damaging (affecting the c2 domain) and that they could also be associated with a higher risk for a malignant lesion in carrier patients [65].

### 4.4. Treatment

There is no specific treatment for the disease, as no international consensus exists on this aspect. Considering the evolutionary course with multiple tumor involvement, the treatment is complicated; practically every tumor form has a specific treatment. Diagnosing and treating each tumor process as early as possible is essential to intervene medically and surgically.

Treatment with inhibitors of the mTOR pathway (Sirolimus), which restore the *PTEN* pathway, is a promising prospect. Since many signs have an early onset [66], mTOR inhibition may represent a suitable chemopreventive strategy to halt Cowden’s disease progression. The prognosis of the disease is unfavorable due to the increased risk of cancer; the correct monitoring of these patients is essential for the survival of these patients. The presented case benefited from surgical treatment only (tonsillectomy, left thyroid lobectomy), with total colectomy and right thyroid lobectomy currently being recommended due to the increased risk of malignancy [67].

*PTEN* loss-of-function mutations, which are oncogenic driver events in CS-related breast cancers, result in dominant AKT activation. Preclinical evidence suggests that cancers with AKT activation have increased sensitivity to AKT inhibition and may improve the outcome of patients with somatic and germline *PTEN* mutations [68]. 

### 4.5. Screening Methods and Genetic Counseling

Monitoring and screening CS patients is challenging for clinicians; currently, opinions are divided. The American College of Gastroenterology Guidelines (A.G.G.) recommend a colonoscopy every two years starting at the age of 15; they also recommend periodic specialist clinical consultations regarding the thyroid, breasts, kidneys, and uterus in the event of the appearance of tumors or the malignant transformation of already present nodules. On the other hand, the NCCN (National Comprehensive Cancer Network) guidelines recommend a colonoscopy from age 35 and subsequent colonoscopies every five years or sooner if polyps are already present. Prophylactic colectomy is also recommended in the presence of multiple polyps with a high risk of malignancy [69]. The screening and prophylactic measures recommended for Cowden syndrome are synthesized in Table 3.


**Screening and prophylaxis recommendations are highlighted in Table 3.**



**Guideline Genturis [70]**


Pathogenic *PTEN* germline mutations are characteristic of *PTEN*-associated hamartomatous syndromes, with different localizations and an increased risk of malignant transformation. Cowden syndrome, an autosomal dominant pathology with high penetrance and the most common *PTEN*-opathy, is often difficult to diagnose due to the variability in its characteristic phenotype. Pathogenic *PTEN* variants associated with excessive cell proliferation and implicit tumorigenesis are transmitted according to the autosomal dominant inheritance pattern, with a 50% recurrence risk for the offspring of an affected parent. Still, approximately 45% of mutations are “de novo” or a consequence of parental mosaicism. Frameshift mutations and loss-of-function truncated proteins appear more commonly in thyroid cancer [60]. The benefits and implications of asymptomatic relatives of a patient diagnosed with CS must be discussed on a case-by-case basis, especially if children are involved. 

## 5. Conclusions

CS is the most prevalent pathology among the syndromes under the umbrella of PHTS. The importance of early diagnosis, along with patient education, genetic counseling, and periodic follow-up, plays a vital role in the evolution and prognosis of this syndrome, with the multidisciplinary team making a significant contribution. The presented case raises awareness regarding the importance of diagnosing this heterogenous syndrome as quickly as possible since, although an early onset, we still need help to establish a diagnosis. 

## Figures and Tables

**Figure 1 genes-14-01909-f001:**
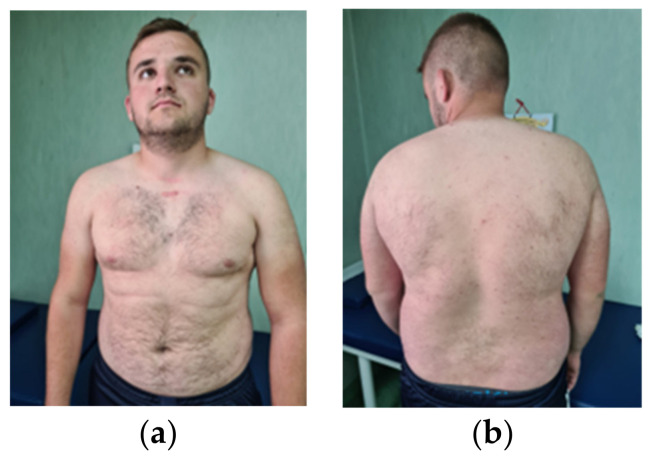
(**a**) Anterior and posterior chest shows multiple pustular lesions; (**b**) keloid scar after thyroid lobectomy; a gynoid pattern of subcutaneous cellular tissue.

**Figure 2 genes-14-01909-f002:**
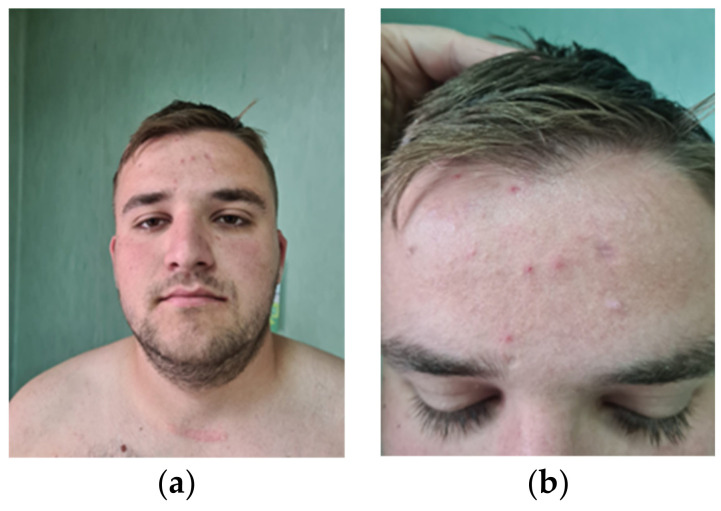
(**a**) Macrocephaly; round face. (**b**) Folliculitis lesions (approx. 2/3 mm scar after removal of a comedo and one skin-colored papillomatous lesion (trichilemmomas) underlying the scar).

**Figure 3 genes-14-01909-f003:**
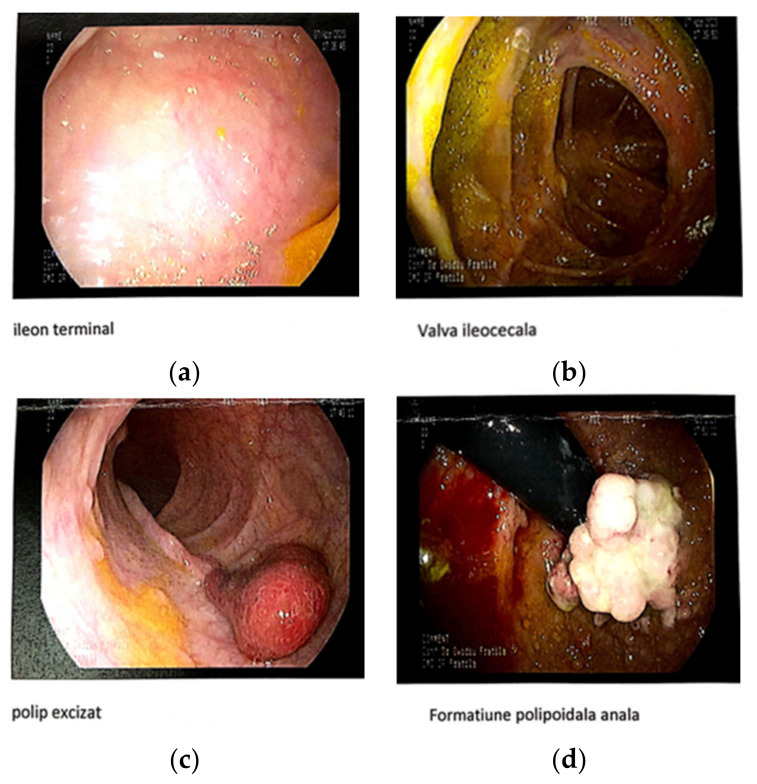
Lower digestive endoscopy: (**a**) terminal ileum; (**b**) ileocecal valve; (**c**) rectal polyp; (**d**) anal polypoid formation.

**Figure 4 genes-14-01909-f004:**
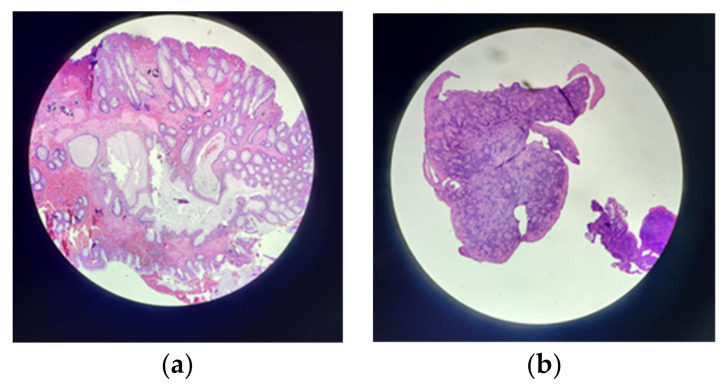
Pathological anatomy: microscopy; (**a**) juvenile polyp; (**b**) anal polypoid formation.

**Table 1 genes-14-01909-t001:** Diagnostic criteria of CS (adapted after the International Cowden Consortium) [12].

**Pathognomonic lesions**: Mucocutaneous lesions: Facial trichilemmomas. Acral keratoses Papillomatous lesions Mucosal lesions		
**Major criteria**	**Minor criteria**	**Observations**
Breast cancer Endometrial cancer Thyroid cancer (mainly follicular thyroid carcinoma) Gastrointestinal hamartomas Lhermitte–Duclos disease, adult Macrocephaly	Other thyroid lesions. Mental retardation (IQ ≤ 75). Gastrointestinal hamartomas. Fibrocystic disease of the breast. Lipomas and fibromas. Genitourinary tumors or malformations.	
**For a person with a negative family history of CS, the following criteria must be taken into account**:		
Pathognomonic mucocutaneous lesions: trichilemmoma or cutaneous facial papules and oral mucosal papillomatosis or oral mucosal papillomatosis and acral keratoses palmoplantar keratoses. Two major criteria. One major and three minor criteria. Four minor criteria.		
**In the case of a person affected by CS, the diagnostic criteria for his relatives are:**		
A pathognomonic mucocutaneous lesion. Any one major criterion with or without minor criteria. Two minor criteria. One of which must be macrocephaly or Lhermitte–Duclos disease.		

**Table 2 genes-14-01909-t002:** Interdisciplinary management.

		Description	Recommendations
ENT (Ear-Nose-Throat)	2006 3 ½ years	Tonsillar polyp, approx. 0.5/0.5 cm, surgically removed.	*Pathological anatomy*: tonsillar papilloma.
	2007 4 years	Tonsillar polyp significantly increased in size, approx. 4/5 cm, associating respiratory distress and oral breathing.	*Tonsillectomy.*
Surgery	2012 9 years	Left plantar hemangioma.	*Plantar ultrasound*: Slightly hypoechoic structure of approx. 6.5/3.1 cm in the continuity of the subcutaneous cellular tissue. *Doppler examination*: Well-represented vasculature with high velocities and extremely variable impedance indices. The appearance suggests plantar hemangioma. *Surgical excision* of the hemangioma.
	2018 15 years	Left thyroid lobe nodules.	*Left thyroid lobectomy.* Pathological examination confirms a follicular adenoma.
	2020 17 years	Contusion wound with partial section of right Achilles tendon.	
Endocrinology	2015 12 years	Left multinodular goiter.	*Thyroid ultrasound*: Right lobe without changes; left lobe: 3 nodular lesions of approx. 3.5/2.7/2 cm, 2.5/2/1.5 cm, and 2/1.5/1.5 cm, respectively.
	2018 15 years	Right thyroid lobe nodule (cyst).	*Ultrasound of the right thyroid lobe*: Average volume, echogenic, homogeneous structure. Inferior, a hyperechoic restricted lesion with minimal proliferation on the anterior wall, non-vascularized cc 0.5 cm (cyst?); vascularization of the thyroid parenchyma of normal appearance.
	2020 17 years	Right thyroid nodule.	
Gastroenterology	2023 April 20 years	Colonic polyposis under observation.	**Lower digestive endoscopy**: Internal hemorrhoids gr II. Suspected inflammatory bowel disease; a colonoscopy was recommended. *Colonoscopy*: At the level of the ileum, colon, and rectum, multiple polyps (hundreds of sessile polyps); sigma: pedunculated polyp with dysplastic appearance—excision; anal canal: polypoid poly-lobed formation of approx. 2.5 cm—biopsy (**Figure 3a–d**). **Pathological anatomy**: Hyperplastic polyps at the level of the rectum; juvenile hamartomatous polyps at the level of the anus (**Figure 4a,b**).
Dermatological	2023, May, 20 years	Facial acne and papular lesion on the forehead with the appearance of trichilemmoma.	A biopsy of the papillomatous lesion was recommended.

**Table 3 genes-14-01909-t003:** Screening methods.

System	Screening and Prophylaxis
Thyroid	From the age of diagnosis (first thyroid cancer reported at 7 years): annual thyroid ultrasound.
Kidneys	Starting at the age of 40, renal ultrasound every 1–2 years. Renal imaging (CT or, preferably, MRI).
Colon	Starting at the age of 35, a colonoscopy every 5 years and more frequently if there are polyps or suggestive symptoms.
Breast	Starting at the age of 30, annual MRI or mammography every 2 years.
Dermatological	An annual dermatological examination is recommended (including dermatoscopies for skin melanoma). Due to the tendency to form keloid scars associated with *PTEN*, the excision of skin lesions is recommended, but only if they show signs of malignancy or generate significant symptoms.
Growth	From the age of diagnosis and at the recommendation of the doctor, psycho-motor evaluations of the child are recommended. Brain M.R.I. scans are recommended if the patient is symptomatic. Patient education and assessment for early intervention as needed.
Genetic counseling	Family genetic counseling. Psychological support is recommended in all cases (e.g., communicating the diagnosis, family planning, prophylactic mastectomy).

## Data Availability

Not applicable.

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
