# Peer review of "A New Frameshift Mutation of PTEN Gene Associated with Cowden Syndrome—Case Report and Brief Review of the Literature"

_genes, 2023, doi:10.3390/genes14101909_

Round 1

Reviewer 1 Report

The authors present a single case with fairly typical PTEN related Cowden syndrome. The case is not unique and finding a novel pathogenic variant does not justify publication on its own. There are not any particularly interesting features to justify publication. The paper is I’m afraid not well put together and is littered with spelling errors and wrong citations

Specific comments

11.       Abstract ‘S. Cowden (CS) is a rare disease in the large group of genodermatoses first described in 1963’ What is ‘S.’? This sentence reads like the large group was first described rather than the specific Cowden phenotype. This sentence needs redoing

22.       ‘Cowden syndrome (MIM≠158350) is a monogenic disease with autosomal dominant transmission, part of PHTS group’ -PHTS should be spelt out first time in main text

33.       ‘skeletal anomalies associated with multiple thyroid adenomas’ -how do thyroid adenomas cause skeletal anomalies?

44.       ‘these patients frequently presenting (with) symptoms from the autistic spectrum, that guides them for seeking medical care.’ -rephrase

55.       ‘Proteus(-like) syndromes characterized by accelerated growth of hamartomatous lesions and body asymmetry but also PTEN-related Proteus syndrome [Ma, H].’ -what is this reference system? I have never come across it and it does not guide the reader to the reference! The references should be numbered

66.       ‘It is a sporadic syndrome with a prevalence of 1:200,000, predominantly in the Caucasian population (96%)’ -What does ‘sporadic’ mean here? Most cases are de novo and therefore isolated. Also it is highly unlikely that it predominates in the Caucasian population with a high de novo rate. It is just that is where it has been reported most

77.        ‘Gastrointestinal involvement is a constant present in CS, the appearance of intestinal polyps being the pathognomonic sign; polyps are present in 35-85% of affected people;’ -35-85% is not constant

88.       ‘Past medical history: From the age of 3 ½ years repeated referrals to different medical specialties all tumor-related:’ -what tumors?

99.       ‘12 years of age a polypomatous lessions of the left thyroid lobe was identified and a left thyroid lobectomy was performed,’ -what was the pathology?

110.   ‘an ultrasound identified a nodular (lession) in the right thyroid lobe that is still under (flllow-up)’ -spelling

111.   ‘20 years of age: a gastroenterology consultation followed by colonoscopy revealed multiple polypomatous (lession) with a suspicion for familial adenomatous polyposis, recommending medical genetic (assement);’ What were the symptoms leading to colonoscopy? What was the appearance? They do not normally appear as sessile polyps typical of FAP and again spelling!

112.   ‘The patient (were) tested using next-generation sequencing (NGS) with a multi-gene panel of 43 genes selected based on associations with Hereditary Gastrointestinal Cancer Panel Plus,’ –‘was’

113.   ‘Phenotypical traits: height 177 cm (75th percentile), weight 97 kg (over 95th percentile); craniofacial dysmorphism: macrocephaly,’ Macrocephaly is insufficient. What was the head circumference and percentile?

114.   ‘The presented case associates macrocephaly with a cranial circumference of 64 cm,’ -This should be in the results

115.   ‘Macrocephaly is found in more than 100 OMIM syndromic or non-syndromic entities like Fragile X, Nevoid Basal Cell Carcinoma (NBCCS), Neurofibromatosis 1’ NF1 is only relative macrocephaly for the patient’s height

116.   ‘The polyps in these syndromes associate an increased risk of colorectal cancer ranging from 9 to 16%’ -in what context? A 9-16% increase in risk is not of concern. I think you mean a 9-16% lifetime risk. This is far less than would be expected if the polyps were predominantly adenoma

117.   ‘In patients with mutations in the PTEN gene, the prevalence of colorectal cancer can reach 90%’ -this is nonsense. This means at any one time 90% of all PTEN carriers will have CRC. The exact quote from the article is ‘. Colorectal polyp prevalence is as high as 90% in patients with PTEN mutations’ -polyps NOT CRC!

118.   ‘In May 2023, following a lower digestive endoscopy and a colonoscopy, more than 100 intestinal polyps were identified from the ileum to the anus. Histopathologically, most were inflammatory.’ -what proportion of all polyps were inflammatory. This is the first time this is mentioned

119.   ‘Without a clear genotype-phenotype correlation, it is suitable for patients diagnosed at a young age with a pathogenic or likely pathogenic variant to be monitored according to the recommendations made by international cancer surveillance guidelines.- reference?

220.   Table 3 is not referenced in the text. Whose recommendations are these? The authors should reference international guidelines such as ERN GENTURIS https://pubmed.ncbi.nlm.nih.gov/32533092/

221.   Breast cancer is not mentioned but this is by far the highest cancer risk in PTEN carriers. This should be in table 3 also

This is very poor and not worthy of publication

Author Response

Esteemed Reviewer, we made all suggested corrections and uploaded a version of the revised manuscript. All changes have been marked as comments to the text.

We also reviewed the Spelling and English Grammar. 

Thank you!

Reviewer 2 Report

The authors presented a case of Cowden syndrome caused by a novel mutation in PTEN gene. I have the following suggestions:

The article is very long, I think the authors should try to make it more precise. For example, the "material and method" and "results" parts can be combined, especially considering this is a case report. 

The quality of English language is fine in general. But grammar mistakes are scattered throughout the article. Many sentences can be shortened to be more precise as well. Recommend careful editing.

Author Response

We reviewed the article and made the changes in the text (cooments to the text)

Round 2

Reviewer 1 Report

I am not impressed with this response. There is no point by point rebuttal. There is not track changes to see what was changed. So I have to search through the manuscript for comments. PHTS was only spelt out in the abstract it should also be spelt out first time in the main text. This article is still not saying anything new and is a straightforward case report with a straightforward pathogenic variant. The authors have long sections on some of the minor tumor associations yet their response to my comment about lack of anything on breast cancer was simply to put a single row in tumor surveillance recommendations

Could still be improved

Author Response

Rebuttal Letter

Case report and a short review of the literature

A New Frameshift Mutation of PTEN Gene Associated with Cowden Syndrome - Case Report and Brief Review of the Literature

The authors present a single case with fairly typical PTEN related Cowden syndrome. The case is not unique and finding a novel pathogenic variant does not justify publication on its own. There are not any particularly interesting features to justify publication. The paper is I’m afraid not well put together and is littered with spelling errors and wrong citations

Specific comments

  1. Abstract ‘S. Cowden (CS) is a rare disease in the large group of genodermatoses first described in 1963’ What is ‘S.’? This sentence reads like the large group was first described rather than the specific Cowden phenotype. This sentence needs redoing

We corrected S with Cowden Syndrome. The sentence was rephrased.

  1. ‘Cowden syndrome (MIM≠158350) is a monogenic disease with autosomal dominant transmission, part of PHTS group’ -PHTS should be spelt out first time in main text

PHTS was spelt in the text.

  1. ‘skeletal anomalies associated with multiple thyroid adenomas’ -how do thyroid adenomas cause skeletal anomalies?

Associated with was changed with and to avoid confusions.

  1. ‘these patients frequently presenting (with) symptoms from the autistic spectrum, that guides them for seeking medical care.’ -rephrase

... which has an earlier onset, these patients frequently present symptoms from the autistic spectrum[1] [3], one of the first signs that alarm and direct parents to specialist medical assessment.

  1. ‘Proteus(-like) syndromes characterized by accelerated growth of hamartomatous lesions and body asymmetry but also PTEN-related Proteus syndrome [Ma, H].’ -what is this reference system? I have never come across it and it does not guide the reader to the reference! The references should be numbered

Proteus syndrome and Proteus(-like) syndromes characterized by accelerated growth of hamartomatous lesions and body asymmetry but also PTEN-related Proteus syndrome.[2]

  1. ‘It is a sporadic syndrome with a prevalence of 1:200,000, predominantly in the Caucasian population (96%)’ -What does ‘sporadic’ mean here? Most cases are de novo and therefore isolated. Also it is highly unlikely that it predominates in the Caucasian population with a high de novo rate. It is just that is where it has been reported most

We rephrased the sentece: Although it has a dominant inheritance pattern, most cases are isolated...

  1. ‘Gastrointestinal involvement is a constant present in CS, the appearance of intestinal polyps being the pathognomonic sign; polyps are present in 35-85% of affected people;’ -35-85% is not constant

Gastrointestinal involvement is common in CS, the appearance of intestinal polyps being the pathognomonic sign; polyps are present in 85% of affected people and have various histologies: hamartomatous, inflammatory, adenomatous, hyperplastic, ganglioneuromatous[3]

  1. ‘Past medical history: From the age of 3 ½ years repeated referrals to different medical specialties all tumor-related:’ -what tumors?

We rephased and further described cronologically the tumors: : From the age of 3 ½ years repeated referrals to different medical specialties all tumor-related as following: 2007 (3 ½ years) a tonsillar papilloma surgically removed at the age 4 ; 2012 at 9 years plantar tumor (diagnoesd as a plantar hemangioma) surgically excised...

  1. ‘12 years of age a polypomatous lessions of the left thyroid lobe was identified and a left thyroid lobectomy was performed,’ -what was the pathology?
  2. ‘an ultrasound identified a nodular (lession) in the right thyroid lobe that is still under (flllow-up)’ -spelling

... at 12 years of age an adenomatous lessions of the left thyroid lobe was identified and a left thyroid lobectomy was performed

  1. ‘20 years of age: a gastroenterology consultation followed by colonoscopy revealed multiple polypomatous (lession) with a suspicion for familial adenomatous polyposis, recommending medical genetic (assement);’ What were the symptoms leading to colonoscopy? What was the appearance? They do not normally appear as sessile polyps typical of FAP and again spelling!

To exclude association with FAP we rephased as following: years of age: the exacerbation of dyspeptic syndrome unresponsive to medication with digestive enzymes and antispasmodics resulted in gastroenterology consultation followed by colonoscopy that revealed multiple polypomatous lession with a suspicion for a genetic-assocaited polyposic synrome...

  1. ‘The patient (were) tested using next-generation sequencing (NGS) with a multi-gene panel of 43 genes selected based on associations with Hereditary Gastrointestinal Cancer Panel Plus,’ –‘was’

Corrected

  1. ‘Phenotypical traits: height 177 cm (75th percentile), weight 97 kg (over 95th percentile); craniofacial dysmorphism: macrocephaly,’ Macrocephaly is insufficient. What was the head circumference and percentile?

macrocephaly with a head circumference of 64 cm (over 99th percentile), ...

  1. ‘The presented case associates macrocephaly with a cranial circumference of 64 cm,’ -This should be in the results
  2. ‘Macrocephaly is found in more than 100 OMIM syndromic or non-syndromic entities like Fragile X, Nevoid Basal Cell Carcinoma (NBCCS), Neurofibromatosis 1’ NF1 is only relative macrocephaly for the patient’s height

We excluded NF1 from the text.

  1. ‘The polyps in these syndromes associate an increased risk of colorectal cancer ranging from 9 to 16%’ -in what context? A 9-16% increase in risk is not of concern. I think you mean a 9-16% lifetime risk. This is far less than would be expected if the polyps were predominantly adenoma

There is a 9 to 16% lifetime risk for malignant transformations for polyps...

  1. ‘In patients with mutations in the PTEN gene, the prevalence of colorectal cancer can reach 90%’ -this is nonsense. This means at any one time 90% of all PTEN carriers will have CRC. The exact quote from the article is ‘. Colorectal polyp prevalence is as high as 90% in patients with PTEN mutations’ -polyps NOT CRC!

Corrected in text

  1. ‘In May 2023, following a lower digestive endoscopy and a colonoscopy, more than 100 intestinal polyps were identified from the ileum to the anus. Histopathologically, most were inflammatory.’ -what proportion of all polyps were inflammatory. This is the first time this is mentioned

...endoscopy and a colonoscopy, more than 100 intestinal polyps were identified from the ileum to the anus. Biopsy was performed for five polyps, four with inflammatory histopathological characteristics, and one adenoma.

  1. ‘Without a clear genotype-phenotype correlation, it is suitable for patients diagnosed at a young age with a pathogenic or likely pathogenic variant to be monitored according to the recommendations made by international cancer surveillance guidelines.- reference?

We added the reference: NCCN guideline

  1. Table 3 is not referenced in the text. Whose recommendations are these? The authors should reference international guidelines such as ERN GENTURIS https://pubmed.ncbi.nlm.nih.gov/32533092/

Cited in text

  1. Breast cancer is not mentioned but this is by far the highest cancer risk in PTEN carriers. This should be in table 3 also

We mentioned breast cancer and added more information related to this pathology.

[1] Lloyd, K. M., 2nd; Dennis, M. Cowden’s Disease. A Possible New Symptom Complex with Multiple System Involvement. Ann. Intern. Med. 196358, 136–142. https://doi.org/10.7326/0003-4819-58-1-136.

[2] Ma, H.; Brosens, L. A. A.; Offerhaus, G. J. A.; Giardiello, F. M.; de Leng, W. W. J.; Montgomery, E. A. Pathology and Genetics of Hereditary Colorectal Cancer. Pathology 201850 (1), 49–59. https://doi.org/10.1016/j.pathol.2017.09.004

[3] Ballester Pilarski R, Burt R, Kohlman W, Pho L, Shannon KM, Swisher E. Cowden syndrome and the PTEN hamartoma tumor syndrome: systematic review and revised diagnostic criteria. J Natl Cancer Inst. 2013 Nov 06;105(21):1607-16.

I am not impressed with this response. There is no point by point rebuttal. There is not track changes to see what was changed. So I have to search through the manuscript for comments. PHTS was only spelt out in the abstract it should also be spelt out first time in the main text. This article is still not saying anything new and is a straightforward case report with a straightforward pathogenic variant. The authors have long sections on some of the minor tumor associations yet their response to my comment about lack of anything on breast cancer was simply to put a single row in tumor surveillance recommendations

First of all, we apologise for not submitting the Rebuttal Letter

Second, we added new information, for a more satisfactory subject approach:

3.4. Molecular investigations

...This variant is absent in gnomAD, a large reference population database (n>120,000 exomes and >15,000 genomes) which aims to exclude individuals with severe pediatric disease. This variant generates a frameshift in exon 7 (of a total of 9 exons) resulting in a premature stop codon. This is predicted to lead to a loss of normal protein function, either through protein truncation or nonsense-mediated mRNA decay. Loss of function is an established disease-mechanism in this gene (HGMD). In silico prediction algorithms (POLYPHEN, SIFT, MUTTASTER)  suport the pathogenicity of the identified variant and therefore the negative effect on the phenotype. To the best of our knowledge, this variant has not been reported in the medical literature or on disease-related variation databases.

4.1.3. Lymphoid involvement. Autoimmune-related phenotypes and lymphoid hyperplasia can be observed in 40% of PHTS patients. These patients show dysregulated immune function with lymphopenia, CD4+ T cell depletion, and changes in T and B lymphocyte subsets.

4.1.4. Gastrointestinal manifestations...Additionally, >90% of PTEN mutation carriers who had a colonoscopy had colorectal polyps, typically with mixed histologies

4.1.6. Cancer risk. Patients with PTEN germline variants have a highre risk for malignant events compared to general population Recent literature data review  reported higher risks for breast cancer (67% to 85%), endometrium cancer (19% to 28%), thyroid cancer (6% to 38%), renal cancer (2% to 24%), colorectal cancer (9% to 32%), and melanoma (0% to 6%) with a Cumulative Lifetime cancer risks (CLTRs) of 36 years of age. Another extensive study revealed a higher incidence for cancer in PTEN carriers, and therefore guide for risk-management strategies with enhanced surveillance approaches with improve clinical outcomes regardless of their initial clinical presentation. Breast cancer is the most common malignancy in PTEN female carriers; tumors are more commonly triple negative (loss of progedsteron. Estrogen and HER2 expression), are more agrresive and could affect both breast. Although a rare entity, breast cancer has also been reported in male PTEN carriers.

4.1.5. Skin changes. Characteristic skin lesions identified in CS include trichilemmomas, acral keratosis, mucocutaneous neuromas, oral papillomas, and macular pigmentation of the gland penis, often the first clues for clinical diagnosis

4.1.7. Neurodevelopmental anomalies have also been reported in PTEN carriers. Along with macrocephaly, research indicates possibly increased rates of developmental delay and autism spectrum disorder (ASD) and other behavioural and psychological manifestations. A represenative procentage of approximately 25% of people with germline PTEN mutations also have characteristic criteria of ASD35. Already mentioned above, macrocephaly is an endophenotipocal trait in CS; interestigely these patients have an increased brain mass and white matter volumes[3] [40]that might be linked to varius intelectual dissabilities and distinctive cognitive profiles (delayed speech, poor working memory and processing speed). Other structural anomalies of the central nervous system like meningiomas, arteriovenous malformations, large perivascular spaces, cortical dysplasia, and gray matter heterotopias have also been reported.  Pathognomonic for CS, Lhermitte-Duclos disease, is diagnosed in adults and is characterisated by a hamartomatous overgrowth of the cerebellum with large neuronal cells expanding into the granular and molecular layers of the cerebellar cortex with a “tiger-striped”appearance on imaginng assesment.

Treatment

…A driver oncogenic event in CS-related breast cancers is PTEN loss-of-function mutations resulting in dominant AKT activation. Preclinical evidence suggests that cancers with AKT activation have increased sensitivity to AKT inhibition and may improve the outcome of patiens with somatic and germline PTEN mutations.

Thank you for your cooments and suggestions.

Round 3

Reviewer 1 Report

This is much improved and the new cancer section is important. There are still issues with the text and in particular the referencing

1.       ‘Phenotypical traits: height 177 cm (75th percentile), weight 97 kg (over 95th percentile); craniofacial dysmorphism: [macrocephaly with a head circumference of 64 cm (over 99th percentile), round face, multiple folliculitis]. -why have you deleted the head circumference?

2.       ‘…with a Cumulative Lifetime cancer risks (CLTRs) of 36 years of age38.’ What is the cumulative risk at 36?

3.       The references require a lot of attention. The authors have introduced references in square brackets but have not deleted the old superscript references. Old reference 10 is deleted but the references are not moved down so there is no new reference 10. The numbering and association with text need very careful checking

4.       The authors have now provided a point by point rebuttal that by and large is satisfactory. However their spelling of the added sections in the rebuttal is appalling although it appears correct in the actual manuscript

5.       Authors need to state that %s in the new cancer section are lifetime risks

Still problems

Author Response

Rebbutal letter

  1. ‘Phenotypical traits: height 177 cm (75th percentile), weight 97 kg (over 95th percentile); craniofacial dysmorphism: [macrocephaly with a head circumference of 64 cm (over 99th percentile), round face, multiple folliculitis]. -why have you deleted the head circumference?

R1.Head circumference was deleted by mistake. We reintroduced the sentence: ...„macrocephaly with a head circumference of 64 cm (over 99th percentile), round face multiple foliculitis lesions”...

  1. ‘…with a Cumulative Lifetime cancer risks (CLTRs) of 36 years of age38.’ What is the cumulative risk at 36?                                                         R2...with a Cumulative Lifetime cancer risks (CLTRs) of 81-90% and a median age at diagnosis of 36 years.
  2. The references require a lot of attention. The authors have introduced references in square brackets but have not deleted the old superscript references. Old reference 10 is deleted but the references are not moved down so there is no new reference 10. The numbering and association with text need very careful checking.

R3. References were checked for accuracy.

  1. The authors have now provided a point by point rebuttal that by and large is satisfactory. However their spelling of the added sections in the rebuttal is appalling although it appears correct in the actual manuscript

R4. We revieed the spelling and revised the errors.

  1. Authors need to state that %s in the new cancer section are lifetime risks

R5. Cumulative Lifetime cancer risks (CLTRs) of 81-90%

Thank you for the comments and suggestions. 
